# Comparison between flaming, mowing and tillage weed control in the vineyard: Effects on plant community, diversity and abundance

Matia Mainardis[☯], Francesco Boscutti[ID]*[☯], Maria del Mar Rubio Cebolla[☯], Gianfranco Pergher

Department of Agricultural, Food, Environmental and Animal Sciences (DI4A), University of Udine, Udine, Italy

☯ These authors contributed equally to this work.
* francesco.boscutti@uniud.it

**Data Availability Statement:** All relevant data are within the manuscript and its Supporting Information files.

## Abstract

The effect of different management techniques for plant control in the vineyard were compared in the present work, focusing on plant diversity preservation and management efficacy in a two-year experiment on vineyard row weed community. Biomass-fueled flame weeding (with two intensities) was applied as an innovative plant control technique in contrast to tillage and mowing practices. The results showed that flaming was comparable to tillage regarding weed control effectiveness, and was more efficient than mowing. However, species number and functional evenness were not substantially modified by changing the applied management technique. Functional trait analysis demonstrated that row management significantly affected the frequency of annual plants, plant height, root depth index, and the occurrence of plants with storage organs. As for species composition, meaningful differences were found: only the two flaming treatments (i.e. gentle vs intense) and the gentle flaming vs mowing had consistent species composition. Flame wedding showed some potential benefits in plant control in the vineyard by favouring small plant and controlling overall weed abundance. On the other hand, flaming favoured plant species with asexual reproduction, with a potential negative impact on weed-vine competition and species persistence in the vineyard. Further studies are required to investigate such contrasting aspects, also considering other weed control techniques (e.g. cover-crops), considering a sustainable perspective of an herbicide-free environment.

## 1. Introduction

The concept of sustainable agriculture involves pesticide reduction as one of the high-priority targets [1]. In this light, thermal weed control is a promising approach aimed at reducing chemical usage in sustainable agricultural schemes [2]. This technique relies on the effects of heat transfer to plant materials (leaves, flowers, stems, propagules), with the aim of destroying cell structures by leading to protein denaturation [3]. The burners, hence, create high

**Funding:** The authors received no specific funding for this work.

**Competing interests:** The authors have declared that no competing interests exist.

temperatures on plants (above lethal levels) with eventual tissue desiccation [4]. Flaming is applied mainly in agriculture and urban areas as primary heat source to control weed in heat-tolerant herbaceous and horticultural crops [5,6], with the aim of reducing chemicals and promoting sustainable practices e.g. in organic farming [7]. Traditional flame weeding involves significant fossil fuel consumption, such as liquefied petroleum gas (LPG) [8,9], propane [10] or diesel oil, but recently an innovative biomass-fueled flaming device (CS Thermos, San Vendemiano, TV, Italy) has been proposed to allow a more sustainable approach to weed control in the vineyards, able to reduce green-house gases (GHG) emissions and fuel costs [11].

Flaming has shown to be effective on weed control, acting differently according to the considered species and its growth-stage [4,12]. In particular, apart from effectiveness in plant removal, weed regrowth is recurrently observed after the death of the plant's above-ground tissues and needs attention to inspect the real treatment efficacy [13]. So far, little is known about the impact of flaming on the whole plant community and its potential selective action on plant species, especially when compared to other weed control techniques (e.g. mowing, tillage, chemicals).

In rainy climates, where weed growth rate is particularly high, weed control between vineyard rows is typically done by mowing or shredding [11]; however, weed control under the rows is critical, due to the difficulty of reaching the vine area and to the possibility of damaging trunks and roots [11]. Mechanical methods for weed control, such as tillage and mowing, are applied with a pool of different machines, commonly equipped with automatic vine-skipping devices to avoid plant damages [14]. Mechanical methods, however, have a number of downsides, including the need of repeated applications (4–6 per year) at low working speeds (2–4 km/h), incomplete weed control around trunks and posts, limitations on wet soils, erosion risks due to tillage in hilly vineyards [11].

Agriculture intensification is considered as one of the main causes of biodiversity loss worldwide [15–17]: as land-use intensification increases, future biodiversity conservation will strongly depend on the capability of cultivated lands to provide suitable habitats for species and communities conservation [18]. Increasing vineyards development in the Mediterranean region is considered as a major driver of conversion of several important habitats [19], targeted for biodiversity conservation by European Union (EU), including Natura2000 sites [18]. Vineyards are recognized as one of the most intensive agricultural forms, resulting in simplified landscapes, where semi-natural vegetation is severely restricted to small and scattered patches [20]. As a consequence, research is stimulating a more sustainable management approach to increase biodiversity in vineyard agro-systems [20–22]. Vine cultivation, in particular, is encountering significant problems in the transition towards more sustainable systems, often harbouring a low plant diversity linked to soil fertility (organic and nutrient content), reduced microbial activity and a general lack of agronomic alternatives that demonstrated to be economically viable [23].

Although several studies reported alternative weed control techniques in the vineyard, so far as we know, no studies analyzed the effect of flaming on plant communities, compared to mechanical weed control techniques. Most of the existing studies evaluated only weed control effectiveness of the different available techniques. In particular, it was shown that type and intensity of management can produce marked changes in the biodiversity associated to the vineyard [20,24]. As an example, concerning the merely weed control, flaming was compared to glyphosate, hot foam and nonanoic acid for weed control, and it was shown that flaming and hot foam were more efficient than the other tested techniques [25]. In this work, the effect of one-sided biomass-fueled flaming at different intensities (slow and fast speed flaming) on plant community composition and abundance in the vineyard was assessed in comparison to mechanical tillage and mowing techniques. We hypothesized that flaming would select certain

species pools (i.e. shift in species composition) and plant functional groups (i.e. changes in functional traits) when compared to tillage and mowing. We also expect flame weeding to reduce the overall plant abundance (cover) and diversity, in particular if compared to mowing.

## 2. Material and methods

### 2.1 Experimental design

The vineyard (cv: Merlot), located in Buttrio (Udine, Italy), was 16 years old and trained to a bi-lateral Guyot, tied to a supporting wire at approx. 1.00 m above the ground [11]. The vines were spaced 1.15 m on the row and 2.80 m between the rows. Until year 2017, the vineyard was managed using sod alleys (i.e., permanent ground cover between the rows) and a combination of tillage and herbicide application to control weeds under the vines (rows) [11]. A field experiment was performed during the growing seasons of years 2018 and 2019 to compare the effect of flaming (with two intensities), tillage and mowing on weed control on the row in the vineyard.

One-sided flaming was applied with a prototype biomass-fueled flamer (CS Thermos, San Vendemiano TV, Italy), consisting of a fuel hopper (capacity: 300 dm$^3$, approx. 200 kg of wood pellet), an auger feeding system, a rotating-grid combustion chamber and a horizontal chimney delivering the flame laterally onto the ground through a curved outlet [11]. Three applications per year were performed, at an average speed respectively of 4.1 km/h (as for slow speed flaming, hereafter intense) and 4.8 km/h (as for fast speed flaming, hereafter gentle) (S1 Table).

The tillage option included one application per year of a disc cultivator (at 4.5 km/h), and two of a weeder blade (at mean 3.2 km/h) (S1 Table). The mowing option consisted of three applications of an undervine mower at 2.4 km/h mean speed (S1 Table). The low working speed assumed here was found necessary in the field tests [11] to allow the cutting discs to penetrate the space between vine trunks, and to reduce the stress on the trunks. Each treatment was applied three times per year during the growing season. The details of the applied management schemes were summarized in S1 Table. It should be highlighted that flaming treatment did not lead to any visible damage to vine trunks, as highlighted also in [11].

The experimental area included 16 test rows (i.e., 4 main treatments x 4 blocks); in each row, 4 sampling points were evenly distributed within the available position between vine plant and vine post, for a total of 64 measures. This choice was made as this position was considered highly representative for weed control efficiency, as one of the most difficult areas to manage [11]. A plastic frame (50 cm x 50 cm) was used as reference measure for each vegetation plot. In 2019, photographic surveys were conducted in Mid-April, start of June and Mid-July, to analyse plant community changes across the season. Photos were taken orthogonally to the plot surface at a constant height of ca. 120 cm above ground. Plant survey was conducted in each period just before the successive treatment (i.e. flaming, tillage, mowing), in order to assess the plant community composition and abundance at the plant growth peaks.

### 2.2 Data collection

Each plant species cover was measured by photo digitalization, using the ImageJ$^®$ software (equipped with ObjectJ$^®$ package). For each photograph, occurring plant species were identified and the area covered by each plant was boarded by polygons. Each species cover area was automatically calculated by setting the photo scale using, as reference, the frame length (50 cm). Taxonomy and nomenclature followed the official list of the Italian flora [26,27]. Prior to analysis, the three seasonal pseudo-replicates were pooled, using the maximum cover value of each plant taxon on a given plot.

Taxonomical diversity was assessed using species richness (i.e. number of species), while functional diversity (FD) was evaluated in terms of functional richness (FRich), evenness (FEve), dispersion (FDis) and divergence (FDiv) [28]. Ten functional traits were considered, i.e. life form, leaf duration, plant height, root depth index, presence of reserve and storage organs, reproduction mode (only seed vs clonal organs), flowering period (number of months), seed mass, leaf area (LA) and specific leaf area (SLA). Traits were derived from 'Flora Indicativa' [29] and "LEDA Traitbase" [30]. These traits were selected as they were expected to be ecologically relevant and respond to major environmental changes [31]. FRich describes the total range of functional trait variability of a community, whereas FEve describes the evenness of the abundance distribution (species cover) in a functional trait space (i.e. a measure of regularity of functional distances). FDis was used as a multidimensional functional diversity index that can be weighted by species abundances [32]. Finally, FDiv shows how the abundance is spread along a functional trait axis, within the range occupied by the whole community [33], in other words, it measures the degree to which the abundance of a community is distributed toward the extremities of the occupied trait space.

The overall plant abundance was assessed as sum of all species cover in each plot and for each sample date, while plant cover standard deviation was calculated as proxy of plant abundance variability. At Mid-July survey, plant cover was compared to plant biomass in order to evaluate an eventual correlation between these two measures. To estimate plant biomass, 64 plots in the same row position were selected close to the monitored plots and successively all the plants occurring within the plastic frame (50 cm x 50 cm) were cut at ground level and collected [11]. All the samples were dried in the oven at 103˚C for 24 h and weighed to determine the dry matter content. A linear model was applied to relate the measured overall plant cover at Mid-July with the measured plant dry matter content, showing that the overall plant cover could be considered as a good proxy of the aboveground weed biomass (S1 Fig).

## 2.3 Statistical analysis

The differences in plant diversity (i.e. taxonomical and functional diversity), plant abundance (i.e. overall cover and standard deviation) and each functional trait among the different management treatments were tested by using linear mixed-effects models (LMM; $p<0.05$) [34]. LMM were applied including the block id (i.e. vineyard row id) as random factor. All the analyses were performed using R statistical Software [35]. LMM were applied using the "nlme" package [36]. Assumptions of models were verified using the diagnostic plots of model residuals. Where model residuals violated any linear model assumption, variables were log-transformed. Pairwise comparisons were performed by least-squares means with Tukey adjustment using the 'lsmeans' R package [37]. Fric and species number showed a high correlation (Pearson's r = 0.77, $p<0.001$). Consequently, only the species number was used in the analysis as proxy of both indices.

The changes in species assemblage among all treatments were inspected using a multivariate approach. A Kruskal's non-metric multidimensional scaling (NMDS) unconstrained ordination [38,39] was performed with Bray-Curtis (dis)similarity index, two dimensions (k = 2) on Wisconsin and square root transformed data (stress = 0.24). The homogeneity of species composition was tested calculating the distance between centroids ('variation' of beta diversity) and testing for homogeneity of multivariate dispersion between treatments. This method produces an independent dissimilarity value for each sample, distance to group centroid [40,41]. The differences in species composition were tested using the PERMANOVA on the distance matrices run with 999 permutations. All the multivariate analyses were performed using the "vegan" R package [42]. Finally, the Indicator Species Analysis was applied to identify

the indicator species for each treatment with the 'labdsv' package [43], run with 999 permutations. The IndVal (Φ) index combines species mean abundance (specificity) and frequencies of occurrences within each group (fidelity) [44,45].

## 3. Results

### 3.1 Plant diversity and abundance

During the survey campaign, an overall number of 18 plant species was found, with a mean of 7 species (min = 3, max = 13) per plot. The most frequently detected taxa were *Taraxacum officinale* (= *Taraxacum* sect. *Taraxacum*) (Frequency = 89.0%), *Trifolium repens* (87.5%), *Chenopodium album* (71.9%), and *Digitaria sanguinalis* (71.9%).

Considering plant diversity indices, functional divergence and distance were significantly affected by vineyard raw management, while species number (representing also functional richness; r = 0.77, p<0.001) and functional evenness were not (Table 1). The functional divergence was significantly higher under mowing than tillage management, while flaming showed intermediate values (Fig 1A). Functional distance exhibited the highest value under intense flaming while decreasing under mowing, gentle flaming and tillage management (Fig 1B).

Vineyard row treatments differently affected the overall abundance of plant community (i.e. plant cover) (Table 1), which also represents the overall plant biomass of the plot ($R^2$ = 0.46, p<0.001; S1 Fig). Plants were more abundant in mowed rows while flamed and tilled rows showed similar values (Fig 1C). The applied management technique (Table 1) did not affect plant cover variability (i.e. standard deviation), even though a slight variability increase was observed in tilled rows.

### 3.2 Response of functional traits to management

Row management significantly affected the frequency of annual plants, plant height, root depth index, and the occurrence of plants with storage organs (Table 2).

The annual plant percentage was higher under tillage treatment, showing similar values for the other management practices (Fig 2A). By analysing plant height, it was seen that plants were smaller under intense flaming than under mowing, gradually increasing under gentle

**Table 1. Results of plant diversity indices.**

| Parameter | | DF | F-value | p-value |
|---|---|---|---|---|
| *Species number* | (Intercept) | 1,57 | 959.17 | <0.0001 |
| | Treatment | 3,57 | 0.82 | 0.489 |
| *Functional evenness* | (Intercept) | 1,57 | 758.23 | <0.0001 |
| | Treatment | 3,57 | 0.16 | 0.920 |
| *Functional divergence* | (Intercept) | 1,57 | 1942.82 | <0.0001 |
| | Treatment | 3,57 | 3.29 | 0.027 |
| *Functional dispersion* | (Intercept) | 1,57 | 984.47 | <0.0001 |
| | Treatment | 3,57 | 8.73 | <0.0001 |
| *Overall plant cover* | (Intercept) | 1,57 | 317.87 | <0.0001 |
| | Treatment | 3,57 | 10.74 | <0.0001 |
| *Plant cover standard deviation* | (Intercept) | 1,57 | 335.11 | <0.0001 |
| | Treatment | 3,57 | 2.18 | 0.100 |

Results of the linear mixed-effects models relating the plant diversity and abundance indices with the applied management treatment. Degrees of freedom (DF), Fisher values (F-value) and p-values are shown.

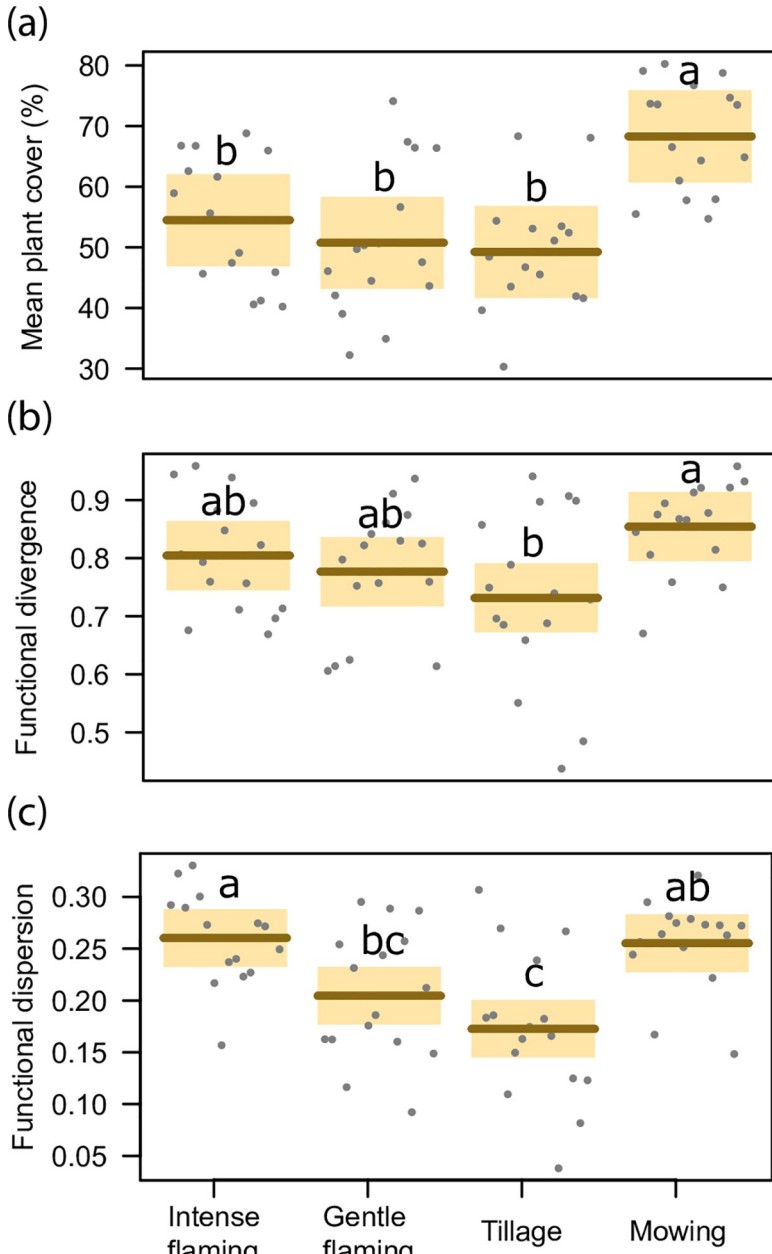

**Fig 1. Plant diversity results.** Comparison between the different applied weed control techniques (intense and gentle flaming, tillage and mowing), including mean plant cover (Fig 1A), functional divergence (Fig 1B) and functional dispersion (Fig 1C). Different letters show significant differences across treatments (p<0.05), while shade areas are the confidence intervals (95%).

flaming and tillage treatments (Fig 2B). Plant mowing advantaged plants with deep roots and presence of storage organs; in contrast, tillage exhibited the lowest values for both traits (Fig 2C and 2D). Flame weeding favoured species with shallow roots but with presence of storage organs.

Row management (Table 2) affected both studied leaf traits. Leaf area was significantly lower in tilled plots, while specific leaf area (SLA) was significantly higher under intense flaming than under mowing (Fig 2E and 2F).

**Table 2. Functional traits results.**

| Parameter | | DF | F-value | p-value |
|---|---|---|---|---|
| Annual species (%) | (Intercept) | 1,57 | 1101.94 | <0.0001 |
| | Treatment | 3,57 | 9.42 | <0.0001 |
| Plant height | (Intercept) | 1,54 | 1966.61 | <0.0001 |
| | Treatment | 3,54 | 3.19 | 0.031 |
| Root depth index | (Intercept) | 1,57 | 12014.07 | <0.0001 |
| | Treatment | 3,57 | 6.08 | 0.001 |
| Species with storage organs(%) | (Intercept) | 1,57 | 1352.29 | <0.0001 |
| | Treatment | 3,57 | 9.53 | <0.0001 |
| Leaf area | (Intercept) | 1,57 | 110.69 | <0.0001 |
| | Treatment | 3,57 | 5.18 | 0.003 |
| Specific leaf area | (Intercept) | 1,57 | 7399.15 | <0.0001 |
| | Treatment | 3,57 | 2.83 | 0.047 |
| Species with reproduction only by seed (%) | (Intercept) | 1,56 | 840.66 | <0.0001 |
| | Treatment | 3,56 | 10.09 | <0.0001 |
| Flowering period length | (Intercept) | 1,57 | 2002.03 | <0.0001 |
| | Treatment | 3,57 | 5.31 | 0.003 |
| Seed mass | (Intercept) | 1,57 | 760.45 | <0.0001 |
| | Treatment | 3,57 | 0.90 | 0.447 |

Results of the linear mixed-effects models relating the mean plant functional traits indices with the applied management treatment. Degrees of freedom (DF), Fisher values (F-value) and p-values are shown.

The type of management affected reproduction strategy and flowering period length but not species seed mass (Table 2). Tillage increased the percentage of species reproducing only by seed and with a longer flowering period, while flaming treatment (both intense and gentle) showed lower and intermediate values, respectively. A longer flowering length generally corresponded to late flowering species, as the correlation between flowering period length and start of flowering period was observed to be high (r = -0.81, p<0.001).

## 3.3 Effects of raw management on species composition

Significant differences were found between species composition under the tested management regimes (PERMANOVA: $R^2$ = 0.15, p = 0.001). The results of pairwise comparison between the applied management techniques showed that only the two flaming treatments (i.e. gentle vs intense) and the gentle flaming vs mowing had consistent species composition (p>0.05). On the other hand, functional traits variation (dispersion) did not differ between the applied treatments (p > 0.05). NMDS highlighted a transition gradient of species composition between the management treatments (Fig 3) and was integrated by the Indicator Species Analysis (ISA), in order to detect the species favoured by each treatment.

NMDS1 discriminated between flaming and the other treatments, while NMDS2 mainly isolated tillage plots from the other management practices. Intense flaming was associated with a high abundance of *Digitaria sanguinalis* (Φ = 0.35; p = 0.08), *Plantago major* (Φ = 0.23; p = 0.16), and *Stellaria media* (Φ = 0.26; p = 0.24). Tillage was characterized by *Veronica persica* (Φ = 0.44; p<0.01), *Polygonum aviculare* (Φ = 0.38; p = 0.02) and *Solanum nigrum* (Φ = 0.23; p = 0.04), while mowing was related to *Chenopodium album* (Φ = 0.37; p = 0.01), *Taraxacum* sect. *Taraxacum* (Φ = 0.38; p = 0.04) and *Sorghum halepense* (Φ = 0.28; p = 0.15). Gentle flaming did not show the occurrence of any indicator species, having a species composition consistent with all the other treatments.

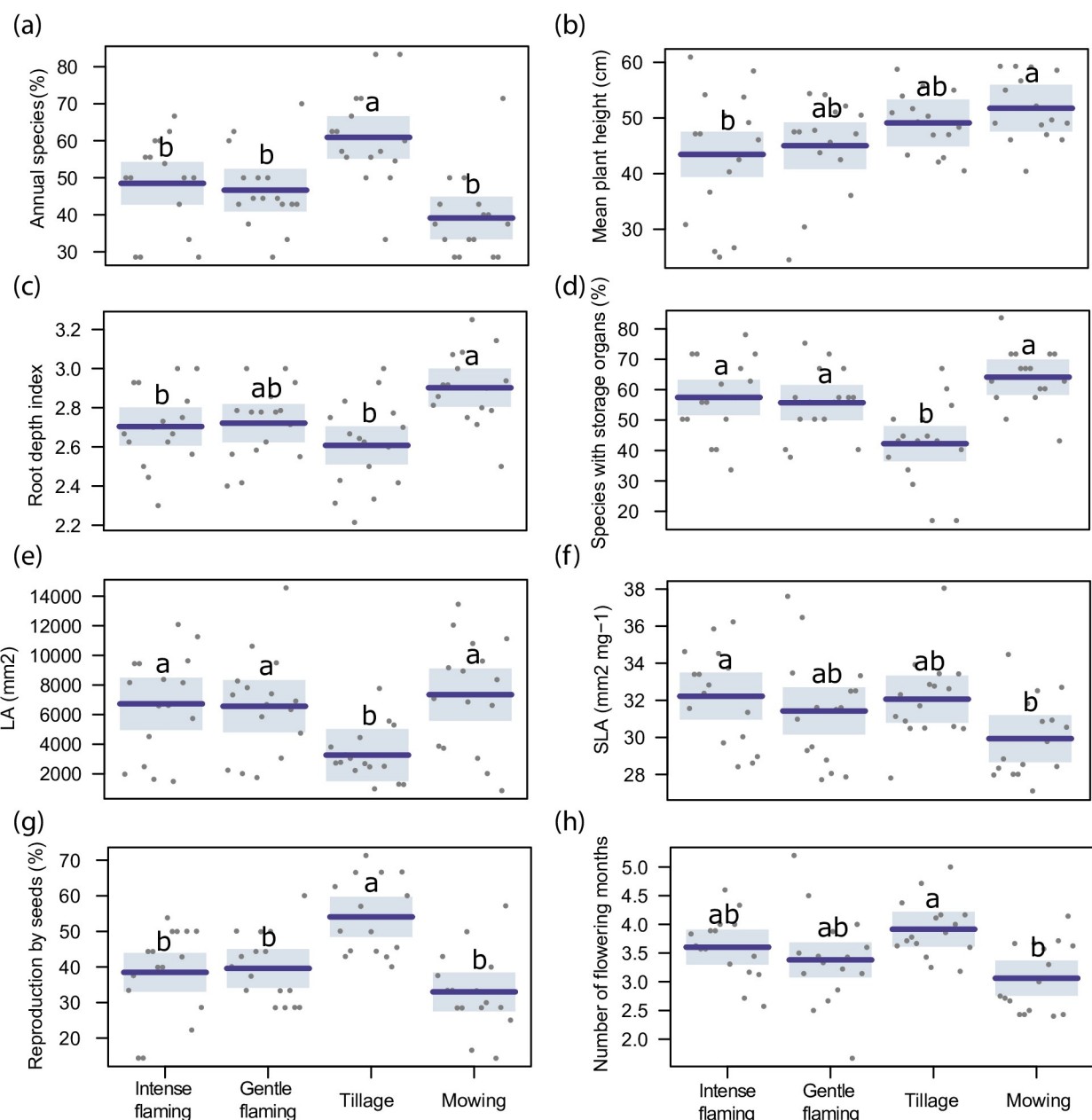

**Fig 2. Functional traits results.** Comparison between the different applied weed control techniques (intense and gentle flaming, tillage and mowing) regarding annual species frequency (a), plant height (b), root depth index (c), presence of storage organs (d), leaf area (e), specific leaf area (f), reproduction by seed (g), number of flowering months (h). Different letters show significant differences across treatments (p<0.05), while shade areas are the confidence intervals (95%).

## 4. Discussion

The actual findings suggest that weed communities in vineyard rows are strongly shaped by the applied management technique, which mainly acts by affecting plant abundance and species composition. The innovative tested biomass-fueled flame weeding showed to contain weed cover (used as a proxy of plant biomass) and plant height, while sustaining the functional

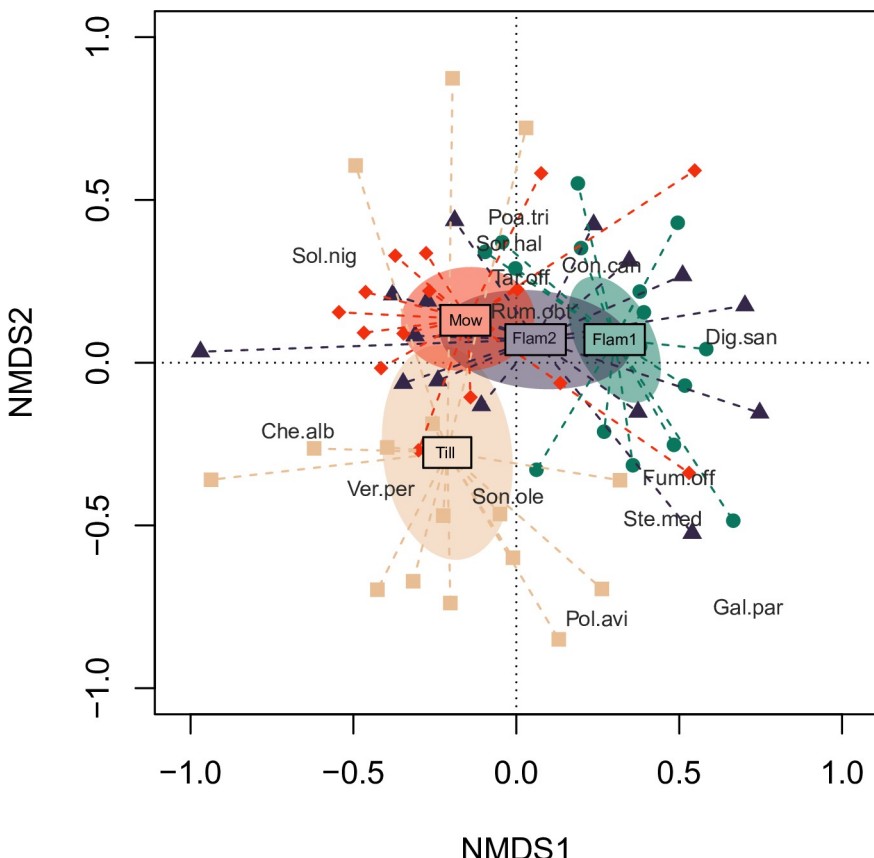

**Fig 3. Non-Metric Multidimensional Scaling (NMDS) ordination.** Centroids of each vineyard row treatment (i.e. Flam1 = intense flaming, Flam2 = gentle flaming, Mow = mowing, Till = tillage) and standard error of the average of scores (shaded elliptic area with 95% confidence limit) are reported. A selection of species was plotted according to species scores and further abundance priority selection (Che.alb = *Chenopodium album*, Con.can = *Conyza canadensis*, Dig.san = *Digitaria sanguinalis*, Fum.off = *Fumaria officinalis*, Gal.par = *Galinsoga parviflora*, Poa.tri = *Poa trivialis*, Pol.avi = *Polygonum aviculare*, Rum.obt = *Rumex obtusifolius*, Sol.nig = *Solanum nigrum*, Son.ole = *Sonchus oleraceus*, Sor.hal = *Sorghum halepense*, Ste.med = *Stellaria media*, Tar.off = *Taraxacum* sect. *Taraxacum*, Ver.per = *Veronica persica*).

diversity in the vineyard. No changes in taxonomical diversity were detected, whereas a meaningful species composition shift was observed between the different management practices, which was also related to changes in functional traits of plant community. However, on the other hand, flaming favored species with an asexual reproduction, negatively impacting weed-crop competition, species persistence in the vineyard and related issues. By adopting a sustainable approach of an herbicide-free environment, additional weed control techniques should be investigated, such as cover-crops.

In addition, the diversification of weed control methods could lead to a more sustainable vineyard management, increasing the ecological diversity of the ecosystem. Recently, innovative weed management was proposed as an ecological friendly approach by combining biological, chemical, cultural and mechanical methods [46]. This strategy consists in increasing the yield and minimizing the economic loss, reducing at the same time the risks for the human health and for the environment and lowering the energy demand [46]. Advanced management techniques include prevention of spreads, cover crops, seedbank management, tillage, crop rotations, biological weed control, and competitive cultivars [46].

## 4.1 Effects of management on plant diversity and abundance

Plant cover was contained under tillage and flame weeding. Respect to mowing, the vegetation underwent a major disturbance which created a difficulty in re-sprouting [25]. Under mowing, instead, where many perennial and taller plants were detected, the plants were ready and adapted for a rapid regrowth (see also discussion about functional groups). Major differences between mowing and flaming treatments were already observed in other ecosystems, but with a scarce effectiveness of burning on weed control [47]. In contrast, our results showed that flame weeding seems an effective treatment in vegetation biomass control.

The functional diversity was higher under mowing and flaming in respect to tillage. Disturbance regime has been proved to be a key driver in promoting functional diversity [48,49]. Tillage is an intensive disturbance action, which results in favoring an homogeneity of functional traits, by selecting particular species guilds [50]. In this light, the present findings confirmed that mowing and flaming are type of disturbances that have relevant effects in sustaining functional heterogeneity. A study about plant communities in four distinct viticulture Regions in Europe already proved that higher management intensity reduced species richness, functional diversity and vegetation cover [51]. Plant functional traits, including the cover of ruderals and annuals, were clearly related to bare soil management; moreover, the type of cover crops influenced the relationship between annual and perennial plants, Grime plant strategy types and species diversity [51].

Unexpectedly, we did not find significant changes in species number between the different applied management techniques. Many authors showed that the management type and intensity are crucial in determining plant diversity [52]. It was highlighted that the different taxa respond in a distinct way to an increase in disturbances, but also to different disturbance types and local site conditions [24]. A meaningful study conducted on vineyards demonstrated that management intensity is a relevant driver in respect to plant diversity: it was seen that high mowing frequency decreased plant diversity [20]. In this light, our findings suggest that a comparable management intensity of diverse techniques produce similar effects on plant diversity.

## 4.2 Changes of functional traits and species composition

The applied management techniques shifted species composition, showing that different pools of species are characteristic of each treatment. Differences in disturbance type produced a change in functional composition of the communities that corresponded to changes in species composition, as other studies confirmed. A large-scale investigation in vineyards in France highlighted that different management practices, such as herbicide application and tillage, promoted some characteristics species pools [52]. Another study [53] investigated the influence of alternative management practices in plant communities composition in South African vineyards, proving that mowing was associated with a higher biodiversity value and a higher cover than tillage and herbicide, promoting at the same time shorter plants, that could be less competitive for grapevines [53]. However, tillage offered an increased plant control efficiency, being beneficial in Summer season, when competition for water may become critical [53].

Our results showed that flaming system favored a peculiar species composition in comparison to mowing and tillage. Consistently, a previous work showed that some species (e.g. *Conyza canadensis*) were more abundant in consequence to a series of flaming treatments [54]. Other authors showed that weed species with unprotected growing points and thin leaves (such as *Chenopodium album*) were more susceptible to flaming than other plants [55].

In this light, we showed that two-year application of flame weeding treatment was sufficient to select a species assembly more resistant to heat stress. As expected, it was found that flame weeding favored smaller perennial plants with shallow root system, often with storage organs,

exhibiting high leaf area and SLA. The reproduction was also integrated by clonal strategies, similar to mowing. Instead, tillage showed a high frequency of annual species reproducing only by seed. Tillage favors annual plants due to a strong soil disturbance which restarts the ecological succession from the soil seed bank, as soil disturbance stimulates massive weed emergence [7,50]. In contrast, flaming and mowing mostly select perennial forms, due to the incomplete destruction of plants and re-sprouting from vegetative organs (e.g. rhizomes, runners, taproots). In agreement, it was proved that plant community response to mowing frequency was mediated by a selection process of resistant growth forms [20].

The plants under flaming and tillage regime were smaller than under mowing (as commented for plant cover abundance species): the applied management selected the plants in order to contain the growth of tall plants. It is plausible that flaming favors plants that have sprouting organs at ground level, where they can better cope flaming destruction. Other studies demonstrated that plant size at treatment time had a major influence on the required flaming intensity [55]. In fact, our results confirmed that intense flaming (i.e. slow application) favors smaller species than gentle treatment. Under flaming, plants have also a shallow root system (plants are smaller in general and hence also root system can be similarly related) but with more storage organs. Storage organs (e.g. swallow roots, rhizomes) allow the plants to take advantage of energy reserves, favoring a rapid re-growth after cutting or burning.

Leaves under flaming were larger but with high SLA. This was an unexpected outcome, because thinner and denser leaves (low SLA) were expected to be more frequent under flaming [55]. In our study, flaming was associated with plants that have basal rosette with large leaves (e.g. *Plantago major*) that can better resist to flaming. Moreover, other literature studies showed that flaming was less effective on plants with succulent leaves (i.e. higher SLA) [56].

Tillage favors species that reproduce only by seed; as demonstrated above, most of them were annual plants. After tillage, soil is bare and species with annual strategy usually produce a large amount of seeds, which populate the soil seed bank. Here the seeds are ready to germinate after each disturbance [7,50]. Under tillage, the plants cover a larger range of flowering periods. These weeds need to be ready to flower and hence produce flowers during the whole growing season (i.e. both in Spring and Summer time).

However, future research in the field is needed to extend the present results, investigating also different soil and climatic conditions, as well as the utilization of alternative herbicide-free weed management solutions in the perspective of an integrated weed management, such as cover-crops.

## 5. Conclusions

In this work, different techniques (intense and gentle flaming, mowing, tillage) were compared to establish weed control effectiveness in vineyards, focusing on the response of plant community and management efficiency, given the environmental importance of cultivated lands. Tillage and flaming were proved to be more efficient than mowing in controlling plants, while species number was substantially unaltered between the different applied techniques. It was confirmed that the analysis of plant functional traits is highly informative in assessing the response of plant communities to weed management practices, especially when compared to usual taxonomical diversity indices. Our findings showed that flaming should be taken into consideration in a sustainable model of vineyard management, as it shifts plant community to a peculiar and favorable composition for an easier management. In addition, flaming maintained plant diversity, contributing to the sustainable agriculture perspective. Mechanical practices were proved to be either less effective (i.e. mowing) or producing high frequency disturbance to soil (i.e. tillage). However, the downsides of this technology have to be carefully

considered, including the impact on weed-crop competition and species persistency in the vineyard.

## Supporting information

**S1 Fig. Correlation between plant biomass and plant cover.** Linear interpolation between measured overall plant cover and measured plant dry matter content at Mid-July.
(PDF)

**S1 Table. Technical details of the applied management techniques.** Application dates and forward speeds of the investigated management techniques in the vineyard (intense and gentle flaming, tillage, mowing).
(DOCX)

**S1 Video. Demonstration of flame weeding application for plant control in the vineyard.**
(MP4)

**S1 Data. Data and species matrix.**
(CSV)

## Acknowledgments

The authors wish to thank Sigismondo Canzian (CS Thermos, San Vendemiano, Italy), Piero Croatto (Vineyard farm, Buttrio, Italy), and Costantino Schincariol (Farmer, Ogliano, Italy) for the precious assistance during the field tests, and Francesco Saccon and Nicola Zucchiatti, for support during field surveys.

## Author Contributions

**Conceptualization:** Francesco Boscutti, Gianfranco Pergher.

**Data curation:** Matia Mainardis, Francesco Boscutti, Maria del Mar Rubio Cebolla, Gianfranco Pergher.

**Formal analysis:** Matia Mainardis, Francesco Boscutti.

**Investigation:** Francesco Boscutti, Maria del Mar Rubio Cebolla, Gianfranco Pergher.

**Supervision:** Gianfranco Pergher.

**Writing – original draft:** Matia Mainardis, Francesco Boscutti.

**Writing – review & editing:** Matia Mainardis, Francesco Boscutti, Gianfranco Pergher.

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
