## [Decision Letter · Decision Letter 0]

1 Jul 2020

PONE-D-20-07695

Innovative versus traditional weed control strategies in the vineyard: flaming affects species composition and abundance but not plant diversity

PLOS ONE

Dear Dr. Boscutti,

Thank you for submitting your manuscript to PLOS ONE. After careful consideration, we feel that it has merit but does not fully meet PLOS ONE’s publication criteria as it currently stands. Therefore, we invite you to submit a revised version of the manuscript that addresses the points raised during the review process.

This manuscript has now been reviewed by two experts. Both reviews agree that the manuscript is well-written and that the approach is interesting, but in their very short review reports they both raised important concerns regarding the title, the focus of the manuscript and the conclusions drawn. I also think that somehow it is not totally clear in the manuscript the novel information that this study brings regarding weed control. Based on this, I would like to give you the opportunity of revising the manuscript taking into account the comments of the reviewers. If you decide to revise the manuscript it will be sent out again to the same or new reviewers for further evaluation. Please provide a detailed response letter with the changes made in the manuscript to take into account the comments.

We look forward to receiving your revised manuscript.

Kind regards,

Amparo Lázaro, PhD

Academic Editor

PLOS ONE

Journal Requirements:

Reviewers' comments:

Reviewer's Responses to Questions

**Comments to the Author**

1. Is the manuscript technically sound, and do the data support the conclusions?

Reviewer #1: Yes

Reviewer #2: Yes

2. Has the statistical analysis been performed appropriately and rigorously? 

Reviewer #1: Yes

Reviewer #2: Yes

3. Have the authors made all data underlying the findings in their manuscript fully available?

Reviewer #1: Yes

Reviewer #2: Yes

4. Is the manuscript presented in an intelligible fashion and written in standard English?

Reviewer #1: Yes

Reviewer #2: Yes

5. Review Comments to the Author

Reviewer #1: The approach is interesting, however there are several serious issues. First of all, practices like mowing are not so...traditional. Secondly, it is almost unbelievable that flaming did not affect plant diversity and number of species. Such finding significantly limits the validity of the data and serious and accurate conclusions cannot be drawn. Therefore, I would modify the title and would not focus on that. Moreover, authors should put considerable more effort to highlight more theri findings and highlight the limited validity of their results and the need of further studies in various soil and climatic conditions.

Reviewer #2: Dear authors, the manuscript is well written with clear material and methods but some points should be revised.

I'm concerned about some points.

- Telling that flaming for weed control is an innovative method: maybe the use of biomass-fueled flaming service is, but not the use o fire for weed control. I strongly recommend title alteration.

- Flaming, mowing and tillage are not the best option for weed control, thinking in a herbicide-free environment.For example, think about cover-crops and possibilities of their use in vineyards. Flaming WAS NOT a win-win choice, as stated in manuscript. The authors should discuss that flaming favored species with assexual reproduction and that this can impact weed-crop competition, species persistence in vineyards and others problems related with it. Also, there is a space to comment about weed control method diversification for non-weed selection.

- In multivariate statistics, it is appropriated to show in figure, NMDS1 and NMDS2 percentages of coverage.

- There is any evaluation of vine plants? There was any effect of fire in plants?

6. PLOS authors have the option to publish the peer review history of their article (what does this mean?). If published, this will include your full peer review and any attached files.

Reviewer #1: No

Reviewer #2: No

---

## [Author Response · Author response to Decision Letter 0]

3 Jul 2020

Editor’s comments

This manuscript has now been reviewed by two experts. Both reviews agree that the manuscript is well-written and that the approach is interesting, but in their very short review reports they both raised important concerns regarding the title, the focus of the manuscript and the conclusions drawn. I also think that somehow it is not totally clear in the manuscript the novel information that this study brings regarding weed control. Based on this, I would like to give you the opportunity of revising the manuscript taking into account the comments of the reviewers. If you decide to revise the manuscript it will be sent out again to the same or new reviewers for further evaluation. Please provide a detailed response letter with the changes made in the manuscript to take into account the comments.

Response: We thank the editor for the opportunity to reconsider our manuscript. We addressed all the comments that arose during the revision process. In particular we changed the title and paid attention to not overstate about our results while highlighting the novelty of our study. For further details, see the below replies to reviewer’s comments. 

Response: The journal’s style was carefully checked and the manuscript was formatted according to the indications. In particular, reference section was modified to fit with the correct style.

Response: Thanks to this kind suggestion, the referred sentence was deleted. All data are already available in the supporting information files. 

Reviewers’ comments

Reviewer #1: The approach is interesting, however there are several serious issues. First of all, practices like mowing are not so...traditional. Secondly, it is almost unbelievable that flaming did not affect plant diversity and number of species. Such finding significantly limits the validity of the data and serious and accurate conclusions cannot be drawn. Therefore, I would modify the title and would not focus on that. Moreover, authors should put considerable more effort to highlight more theri findings and highlight the limited validity of their results and the need of further studies in various soil and climatic conditions.

Response: The title was completely modified according to this kind observation. The need for further studies in the field was highlighted in the revised manuscript version, together with pointing out that the present obtained results are only a single case-study and consequently cannot be extended to other situations without an in depth experimental campaign.

Reviewer #2: Dear authors, the manuscript is well written with clear material and methods but some points should be revised.

I'm concerned about some points.

- Telling that flaming for weed control is an innovative method: maybe the use of biomass-fueled flaming service is, but not the use o fire for weed control. I strongly recommend title alteration.

Response: The title was modified according to this observation and also considering the remarks of other reviewers.

- Flaming, mowing and tillage are not the best option for weed control, thinking in a herbicide-free environment. For example, think about cover-crops and possibilities of their use in vineyards. Flaming WAS NOT a win-win choice, as stated in manuscript. The authors should discuss that flaming favored species with assexual reproduction and that this can impact weed-crop competition, species persistence in vineyards and others problems related with it. Also, there is a space to comment about weed control method diversification for non-weed selection.

Response: Thanks for this interesting suggestion. The discussion was enhanced with relevant and recent literature studies, considering also the negative aspects of flaming treatment and suggesting different techniques for an herbicide-free weed control in the winery with a sustainable perspective.

- In multivariate statistics, it is appropriated to show in figure, NMDS1 and NMDS2 percentages of coverage.

Response: Actually according to main references, contrary to other ordination methods (e.g. PCA, PCoA, CA) which are eigenvector-based methods, NMDS calculations do not maximize the variability associated with individual axes of the ordination (Legendre, P., & Legendre, L. F. – 2012 . Numerical ecology. Elsevier). For this reason there is no % of variance associated with each NMDS axis.

- There is any evaluation of vine plants? There was any effect of fire in plants?

Response: No damages were observed on vine plants after flaming treatment. This observation was added to the revised manuscript version.

---

## [Decision Letter · Decision Letter 1]

27 Jul 2020

PONE-D-20-07695R1

Comparison between flaming, mowing and tillage weed control in the vineyard: effects on plant community, diversity and abundance

PLOS ONE

Dear Dr. Boscutti,

Thank you for submitting your manuscript to PLOS ONE. After careful consideration, we feel that it has merit but does not fully meet PLOS ONE’s publication criteria as it currently stands. Therefore, we invite you to submit a revised version of the manuscript that addresses the points raised during the review process.

ACADEMIC EDITOR: The revised manuscript has now been reviewed by one of the previous reviewers. The reviewer considers that the authors have addressed previous comments adequately, but have several concerns regarding the conclusions of the study. And I concur with these comments. Please, revise carefully the conclusions of the study and submit a revised manuscript together with a response letter answering point-by-point the comments raised.

We look forward to receiving your revised manuscript.

Kind regards,

Amparo Lázaro, PhD

Academic Editor

PLOS ONE

Reviewers' comments:

Reviewer's Responses to Questions

**Comments to the Author**

1. If the authors have adequately addressed your comments raised in a previous round of review and you feel that this manuscript is now acceptable for publication, you may indicate that here to bypass the “Comments to the Author” section, enter your conflict of interest statement in the “Confidential to Editor” section, and submit your "Accept" recommendation.

Reviewer #2: All comments have been addressed

2. Is the manuscript technically sound, and do the data support the conclusions?

Reviewer #2: No

3. Has the statistical analysis been performed appropriately and rigorously? 

Reviewer #2: Yes

4. Have the authors made all data underlying the findings in their manuscript fully available?

Reviewer #2: Yes

5. Is the manuscript presented in an intelligible fashion and written in standard English?

Reviewer #2: Yes

6. Review Comments to the Author

Reviewer #2: First, thanks for the authors modification on manuscript. Ir sounds better about flaming as a weed control method.

But, before it publication, some chances need to be done in conclusions.

Follow their conclusion and observations

In this work, different techniques (intense and gentle flaming, mowing, tillage) were applied to control weeds in vineyards with an environmental-friendly approach, reducing chemical consumption and boosting for an increased plant diversity, give the ecological importance of cultivated lands. ---- For me both are not environment-friendly weed control. Also its not a conclusion to be here.

As a novelty we found the analysis of plant functional trait to be highly informative in assessing the response of plant communities to weed management practices especially when compared to usual taxonomical diversity indices. ---- For me not a novelty, lot of articles found that before, WE CONFIRM.

Our findings showed that flaming could be considered an appropriate practice in a sustainable model of vineyard management, aiming at preservation of economic and environmental resources ---- For me authors did not analysed preservation of economic and environmental resources. Just state that flaming should be taken into consideration as a weed control technique as is shifts community.

In addition, flaming maintained plant diversity, contributing to sustainable agriculture perspective. However, the downsides of this technology have to be carefully considered, including the impact on weed-crop competition and species persistency in the vineyard. Mechanical practices were proved to be either less effective (i.e. mowing) or producing high frequency disturbance to soil (i.e. tillage). ---- OK for me

Future research in the field is needed to extend the present results, investigating also different soil and climatic conditions, as well as the utilization of alternative herbicide-free weed management solutions in the perspective of an integrated weed management, such as cover-crops. --- This is discussion not conclusion

7. PLOS authors have the option to publish the peer review history of their article (what does this mean?). If published, this will include your full peer review and any attached files.

Reviewer #2: No

---

## [Author Response · Author response to Decision Letter 1]

28 Jul 2020

ACADEMIC EDITOR: The revised manuscript has now been reviewed by one of the previous reviewers. The reviewer considers that the authors have addressed previous comments adequately, but have several concerns regarding the conclusions of the study. And I concur with these comments. Please, revise carefully the conclusions of the study and submit a revised manuscript together with a response letter answering point-by-point the comments raised.

Response: We thank the editor for the opportunity to reconsider our manuscript. We addressed all the comments about the conclusion arose by reviewer 2.

Reviewer #2: First, thanks for the authors modification on manuscript. Ir sounds better about flaming as a weed control method.

But, before it publication, some chances need to be done in conclusions.

Response: The conclusion section was revised in accordance to reviewer comments, as kindly requested.

Follow their conclusion and observations

REVIEWER

In this work, different techniques (intense and gentle flaming, mowing, tillage) were applied to control weeds in vineyards with an environmental-friendly approach, reducing chemical consumption and boosting for an increased plant diversity, give the ecological importance of cultivated lands. ---- For me both are not environment-friendly weed control. Also its not a conclusion to be here.

Response: The sentence was modified considering reviewer’s observations, focusing on the conclusions drawn from the experimental design.

As a novelty we found the analysis of plant functional trait to be highly informative in assessing the response of plant communities to weed management practices especially when compared to usual taxonomical diversity indices. ---- For me not a novelty, lot of articles found that before, WE CONFIRM.

Response: The referred sentence was modified as kindly suggested.

Our findings showed that flaming could be considered an appropriate practice in a sustainable model of vineyard management, aiming at preservation of economic and environmental resources ---- For me authors did not analysed preservation of economic and environmental resources. Just state that flaming should be taken into consideration as a weed control technique as is shifts community.

Response: The reference to “economic and environmental resources” was removed and the observation to the shift in plant community composition was added.

In addition, flaming maintained plant diversity, contributing to sustainable agriculture perspective. However, the downsides of this technology have to be carefully considered, including the impact on weed-crop competition and species persistency in the vineyard. Mechanical practices were proved to be either less effective (i.e. mowing) or producing high frequency disturbance to soil (i.e. tillage). ---- OK for me

Future research in the field is needed to extend the present results, investigating also different soil and climatic conditions, as well as the utilization of alternative herbicide-free weed management solutions in the perspective of an integrated weed management, such as cover-crops. --- This is discussion not conclusion

Response: The referred sentence was moved to the end of discussion section, as kindly required. In addition, some further details about the effectiveness of the proposed management techniques were included in the Conclusions section.

---

## [Editor Report · Decision Letter 2]

17 Aug 2020

Comparison between flaming, mowing and tillage weed control in the vineyard: effects on plant community, diversity and abundance

PONE-D-20-07695R2

Dear Dr. Boscutti,

We’re pleased to inform you that your manuscript has been judged scientifically suitable for publication and will be formally accepted for publication once it meets all outstanding technical requirements.

Kind regards,

Amparo Lázaro, PhD

Academic Editor

PLOS ONE
---

## [Editor Report · Acceptance letter]

19 Aug 2020

PONE-D-20-07695R2 

Comparison between flaming, mowing and tillage weed control in the vineyard: effects on plant community, diversity and abundance 

Dear Dr. Boscutti:

I'm pleased to inform you that your manuscript has been deemed suitable for publication in PLOS ONE. Congratulations! Your manuscript is now with our production department. 

Kind regards, 

on behalf of

Dr. Amparo Lázaro 

Academic Editor

PLOS ONE